# Towards Inferential Reproducibility of Machine Learning Research

**Michael Hagmann[1], Philipp Meier[1], Stefan Riezler[1,2]**
Computational Linguistics[1] & IWR[2]
Heidelberg University, Germany
`{hagmann,meier,riezler}@cl.uni-heidelberg.de`

## Abstract

Reliability of machine learning evaluation — the consistency of observed evaluation scores across replicated model training runs — is affected by several sources of nondeterminism which can be regarded as measurement noise. Current tendencies to remove noise in order to enforce reproducibility of research results neglect inherent nondeterminism at the implementation level and disregard crucial interaction effects between algorithmic noise factors and data properties. This limits the scope of conclusions that can be drawn from such experiments. Instead of removing noise, we propose to incorporate several sources of variance, including their interaction with data properties, into an analysis of significance and reliability of machine learning evaluation, with the aim to draw inferences beyond particular instances of trained models. We show how to use linear mixed effects models (LMEMs) to analyze performance evaluation scores, and to conduct statistical inference with a generalized likelihood ratio test (GLRT). This allows us to incorporate arbitrary sources of noise like meta-parameter variations into statistical significance testing, and to assess performance differences conditional on data properties. Furthermore, a variance component analysis (VCA) enables the analysis of the contribution of noise sources to overall variance and the computation of a reliability coefficient by the ratio of substantial to total variance.

## 1 Introduction

Training of deep learning models utilizes randomness to improve generalization and training efficiency, thus causing an inherent nondeterminism that hampers the reliability of machine learning evaluation — the consistency of the measurement of evaluation scores across replicated training runs. Gundersen et al. (2022) list several sources of nondeterminism, e.g., implementation-level nondeterminism such as random ordering in floating-point accumulation in parallel GPU threads (Pham et al., 2021), algorithmic factors such as variations in meta-parameters and model architecture (Lucic et al., 2018; Henderson et al., 2018; D'Amour et al., 2020), or data-level factors such as variations in pre-processing and evaluation metrics (Post, 2018; Chen et al., 2022) or varying characteristics of data in different splits (Gorman & Bedrick, 2019; Søgaard et al., 2021). Zhuang et al. (2022) show that implementation-level nondeterminism is partly irreducible, leading to variability in evaluation scores even for training runs on identical data, algorithmic settings and infrastructure. Furthermore, they point out strong effects of certain types of algorithm-level nondeterminism on certain subsets of the data.

Regarding the comparison of machine learning models, minor variations in these sources of nondeterminism can have huge impact on the resulting evaluation scores and sometimes even reverse the relation between optimal results for baseline and state-of-the-art (SOTA) model (Reimers & Gurevych, 2017; Melis et al., 2018). This fact questions what can be validly learned from a typical machine learning experiment. One current answer is to foster training reproducibility[1] in the sense of an exact duplication of a state-of-the-art (SOTA) training result under exactly the same conditions. In this view, all sources of nondeterminism are regarded as noise or nuisance factors

---

[1]The term was coined by Leventi-Peetz & Östreich (2022) and corresponds to Drummond (2009)'s replicability.

(Forde & Paganini, 2019) that are independent of the learning signal and need to be removed (or at least reduced, even if incurring a cost in efficiency (Ahn et al., 2022)). This goal is pursued by enforcing open-source program code, publicly available data, and explicit descriptions of experimental settings, following reproducibility checklists (Heil et al., 2021; Pineau et al., 2021; Lucic et al., 2022). An unintended side effect of this approach is that the conclusions that can be drawn from such experiments are restricted to statements about a single training configuration on a single test set.

Another viewpoint is to embrace certain types of nondeterminism as inherent and irreducible conditions of measurement that contribute to variance in performance evaluation in an interesting way. Instead of attempting to remove them, we propose to analyze the various components of measurement noise, and especially their interactions with certain properties of data. Such a study can be seen to fall under the umbrella of inferential reproducibility.[2] Goodman et al. (2016) define it to refer to the drawing of qualitatively similar conclusions from either an independent replication of a study or a reanalysis of the original study. For the case of machine learning evaluation, we focus on algorithmic-level factors such as variations in meta-parameters and model architecture and on data-level factors as the main sources of non-determinism in training replications. These are usually described independently of each other. Our goal is to answer the question whether a competitor model yields improvements over a baseline across different meta-parameter settings and across different characteristics of input data, and how variations in algorithmic settings interact with varying data characteristics.

The main contribution of our paper is to show how to apply well-known statistical methods to analyze machine learning evaluations under variability in meta-parameter settings and dependent on data characteristics, with a special focus on the detection of sources of variance and their interaction with data properties. These methods are based on *linear mixed effects models (LMEMs)* fitted to performance evaluation scores of machine learning algorithms. First, we conduct a *generalized likelihood ratio test (GLRT)* to assess statistical significance of performance differences between algorithms, while simultaneously acknowledging for variation in nondeterministic factors. While applicable to any source of nondeterminism, in this paper, we focus on meta-parameter variations as main source of variability. A key feature of our approach is the possibility to assess the significance performance differences under meta-parameter variation conditional on data properties. Second, we show how to use *variance component analysis (VCA)* to facilitate a nuanced quantitative assessment of the sources of variation in performance estimates. Lastly, we compute a *reliability coefficient* to assess the general robustness of the model by the ratio of substantial variance out of total variance. Reliability is also intimately related to the power of the significance test.

Code (R and Python) for the toolkit and sample applications are publicly available.[3]

## 2 Linear Mixed Effects Models

A linear mixed effects model (LMEM) is an extension of a standard linear model that allows a rich linear structure in the random component of the model, where effects other than those that can be observed exhaustively (so-called *fixed effects*) are treated as a random samples from a larger population of normally distributed random variables (so-called *random effects*).

The general form of an LMEM is

$$\mathbf{Y} = \mathbf{X}\boldsymbol{\beta} + \mathbf{Z}\mathbf{b} + \boldsymbol{\epsilon}, \tag{1}$$

where $\mathbf{X}$ is an $(N \times k)$-matrix and $\mathbf{Z}$ is an $(N \times m)$-matrix, called model- or design-matrices, which relate the unobserved vectors $\boldsymbol{\beta}$ and $\mathbf{b}$ to $\mathbf{Y}$. $\boldsymbol{\beta}$ is a $k$-vector of fixed effects and $\mathbf{b}$ is an $m$-dimensional random vector called the random effects vector. $\boldsymbol{\epsilon}$ is an $N$-dimensional vector called the error component. The random vectors are assumed to have the following distributions:

$$\mathbf{b} \sim \mathcal{N}(0, \psi_\theta), \ \boldsymbol{\epsilon} \sim \mathcal{N}(0, \boldsymbol{\Lambda}_\theta), \tag{2}$$

---

[2]This term corresponds to Drummond (2009)'s reproducibility and was coined by Goodman et al. (2016). Instead of contributing further to the terminological confusion in this area, we refer to the brief history of this discussion in Plesser (2018).

[3]https://www.cl.uni-heidelberg.de/statnlpgroup/empirical_methods_tutorial/

where $\psi_\theta$ and $\mathbf{\Lambda}_\theta$ are covariance matrices parameterized by the vector $\theta$.

The most common application of LMEMs is to model complex covariance structures in the data when the usual i.i.d. assumptions fail to be applicable. This is the case for repeated or grouped, and thus non-independent, measurements such as multiple ratings of same items by same subjects in psycho-linguistic experiments. LMEMs have become popular in this area due to their flexibility (Baayen et al., 2008; Bates et al., 2015), and have even been credited as candidates to replace ANOVA (Barr et al., 2013) to analyze experimental data. The price for this flexibility is an elaborate estimation methodology for which we refer the reader to Appendix A2 of Riezler & Hagmann (2022) and to further literature (Pinheiro & Bates, 2000; McCulloch & Searle, 2001; West et al., 2007; Demidenko, 2013; Wood, 2017).

# 3 GENERALIZED LIKELIHOOD RATIO TESTS W/ AND W/O MEASUREMENT VARIATION

Let us assume our goal is to test the statistical significance of an observed performance difference between a baseline and a SOTA system. Furthermore, let us assume we are comparing Natural Language Processing (NLP) models on a benchmark test set of gold standard sentences. In order to conduct a generalized likelihood ratio test (GLRT) for this purpose, we need to fit two LMEMs on the performance evaluation data of baseline and SOTA system which analyze the data differently, and compare their likelihood ratio. Let us further assume an experimental design where variants of the baseline and SOTA models, corresponding to different meta-parameter configurations during training, are evaluated on the benchmark data. Simple linear models are a suboptimal choice to analyze this experiment since they are based on the assumption that each system was evaluated once on a disjoint set of sentences. This would force us to average over variants, thereby losing useful information contained in the clusters of repeated measurements of the same test input.

LMEMs allow us to better reflect this design and to leverage its statistical benefits by adding a random effect $b_s$ for each sentence in our evaluation model. Such a model decomposes the total variance of the evaluation score into three blocks: systematic variance due to the fixed effects of the model, variance due to sentence heterogeneity, and unexplained residual variance. This allows us to reduce the as of yet unaccounted residual variance by attributing a variance component $\sigma_s^2$ to variance between sentences. If we think of the residual error as noise that masks the signal of measured performance scores, we can effectively perform a noise reduction that increases the power of our tests to detect significant differences.

A straightforward technique to implement statistical significance tests using LMEMs is the so-called *nested models* setup (Pinheiro & Bates, 2000). First we train an LMEM that doesn't distinguish between systems. This restricted model

$$m_0 : Y = \beta + b_s + \epsilon_{res} \tag{3}$$

specifies a common mean $\beta$ for both systems as fixed effect, and a sentence-specific deviation $b_s$ as random effect with variance $\sigma_s^2$, and a residual error $\epsilon_{res}$ with variance $\sigma_{res}^2$ for the performance scores $Y$. It represents the null hypothesis that there is no difference between systems. This model is compared to a more general model that allows different means for baseline and SOTA scores:

$$m_1 : Y = \beta + \beta_c \cdot \mathbb{I}_c + b_s + \epsilon_{res} \tag{4}$$

This model includes an indicator function $\mathbb{I}_c$ to activate a fixed effect $\beta_c$ that represents the deviation of the competing SOTA model from the baseline mean $\beta$ when the data point was obtained by a SOTA evaluation. The restricted model is a special case of this model (thus "nested" within the more general model) since it can be obtained by setting $\beta_c$ to zero. Let $\ell_0$ be the likelihood of the restricted model, and $\ell_1$ be the likelihood of the more general model, the intuition of the likelihood ratio test is to reject the null hypothesis of no difference between systems if the statistic

$$\lambda = \frac{\ell_o}{\ell_1} \tag{5}$$

yields values close to zero.

The incorporation of a random sentence effect $b_s$ introduces a pairing of systems on the sentence level that corresponds to standard pairwise significance tests. However, clustering at the sentence

level allows accounting for arbitrary kinds of uncertainty introduced by the random nature of the training process. This setup is thus not only suitable for pairwise comparisons of best baseline and best SOTA model in order to test training reproducibility, but it also allows incorporating broader variations induced by meta-parameter settings of baseline and SOTA systems, thus making it suitable to test inferential reproducibility.

A further distinctive advantage of GLRTs based on LMEMs is that this framework allows analyzing significance of system differences conditional on data properties. For example, we could extend models $m_0$ and $m_1$ by a fixed effect $\beta_d$ modeling a test data property $d$ like readability of an NLP input sequence, or rarity of the words in an input sequence, and by an interaction effect $\beta_{cd}$ allowing to assess the expected system performance for different levels of $d$. The enhanced model

$$m'_1 : Y = \beta + \beta_d \cdot d + (\beta_c + \beta_{cd} \cdot d) \cdot \mathbb{I}_c + b_s + \epsilon_{res} \tag{6}$$

would then be compared to a null hypothesis model of the form

$$m'_0 : Y = \beta + \beta_d \cdot d + b_s + \epsilon_{res}. \tag{7}$$

GLRTs belong to the oldest techniques in statistics, dating back to Neyman & Pearson (1933); Wilks (1938). For more information on extensions of GLRTs for multiple comparisons and on their asymptotic statistics we refer the reader to Chapter 4 and Appendix A3 of Riezler & Hagmann (2022) and to further literature (van der Vaart, 1998; Pinheiro & Bates, 2000; Pawitan, 2001; Davison, 2003; Larsen & Marx, 2012).

## 4 VARIANCE COMPONENT ANALYSIS AND RELIABILITY COEFFICIENTS

The main goal of a reliability analysis in the context of a reproducibility study is to quantify and analyze the sources of randomness and variability in performance evaluation, and to quantify the robustness of a model in a way that allows to draw conclusions beyond the concrete experiment. The first goal can be achieved by performing a variance component analysis (VCA). For example, let us assume we want to specify a model for performance evaluation scores that besides a global mean $\mu$ specifies random effects to account for variations in the outcome $Y$ specific to different sentences $s$ and specific to different settings of a regularization parameter $r$. A tautological decomposition of the response variable into the following four components can be motivated by classical ANOVA theory (Searle et al., 1992; Brennan, 2001):

$$Y = \mu + (\mu_s - \mu) + (\mu_r - \mu) + (Y - \mu_s - \mu_r + \mu). \tag{8}$$

The components of the observed score $Y$ for a particular regularization setting $r$ on a single sentence $s$ are the grand mean $\mu$ of the observed evaluation score across all levels of regularization and sentences; the deviation $\nu_s = (\mu_s - \mu)$ of the mean score $\mu_s$ for a sentence $s$ from the grand mean $\mu$; the deviation $\nu_r = (\mu_r - \mu)$ of the mean score $\mu_r$ for a regularization setting $r$ from the grand mean $\mu$; and the residual error, reflecting the deviation of the observed score $Y$ from what would be expected given the first three terms. Except for $\mu$, each of the components of the observed score varies from one sentence to another, from one regularization setting to another, and from one regularization-sentence combination to another. Since these components are uncorrelated with each other, the total variance $\sigma^2(Y - \mu)$ can be decomposed into the following *variance components*:

$$\sigma^2(Y - \mu) = \sigma_s^2 + \sigma_r^2 + \sigma_{res}^2, \tag{9}$$

where $\sigma_s^2$ and $\sigma_r^2$ denote the variance due to sentences and regularization settings, and $\sigma_{res}^2$ denotes the residual variance component including the variance due to interaction of $s$ and $r$.

Let $\nu_f = \mu_f - \mu$ denote a deviation from the mean for a facet[4] $f$ whose contribution to variance we are interested in. Instead of estimating the corresponding variance components $\sigma_f$ by ANOVA expected mean square equations, we use LMEMs to model each $\nu_f$ as a component of the random

---

[4]In the psychometric approach of Brennan (2001), the conditions of measurement that contribute to variance in the measurement besides the objects of interest are called *facets* of measurement. Facets comprise what we called measurement noise above. In our running NLP example, the objects of interest in our measurement procedure are the sentences. They are the essential conditions of measurement. The only facet of measurement in this example are the regularization settings, while the objects of interest are not usually called a facet.

effects vector $\mathbf{b}$ in equation 2, and model each corresponding variance component $\sigma_f^2$ as an entry of the diagonal variance-covariance matrix $\psi_\theta$ in equation 2.

Besides greater flexibility in estimation[5], LMEMs also allow analyzing the interaction of meta-parameters and data properties. This can be achieved, for example, by changing the random effect $b_r$ to a fixed effect $\beta_r$, and by adding a fixed effect $\beta_d$ modeling test data characteristics, and an interaction effect $\beta_{rd}$ modeling the interaction between data property $d$ and meta-parameter $r$.

The final ingredient of a reliability analysis is the definition of a coefficient that relates variance components to each other, instead of inspecting them in isolation. The key concept is the so-called *intra-class correlation coefficient (ICC)*, dating back to Fisher (1925). A fundamental interpretation of the ICC is as a measure of the proportion of variance that is attributable to substantial variance, i.e., to variance between the objects of measurement. The name of the coefficient is derived from the goal of measuring how strongly objects in the same class are grouped together in a measurement. Following Brennan (2001), we can define a concrete reliability coefficient, denoted by $\varphi$, for our application scenario. In our case, objects of interest are test sentences $s$, and substantial variance is variance $\sigma_s^2$ between sentences. Assume facets $f_1, f_2, \ldots$ and selected interactions $sf_1, sf_2, f_1f_2, \ldots$. Then the reliability coefficient $\varphi$ is computed by the ratio of substantial variance $\sigma_s^2$ to the total variance, i.e., to itself and the error variance $\sigma_\Delta^2$ that includes variance components for all random effects and selected interactions of random effects:

$$\varphi = \frac{\sigma_s^2}{\sigma_s^2 + \sigma_\Delta^2}, \tag{10}$$

where $\sigma_\Delta^2 = \sigma_{f_1}^2 + \sigma_{f_2}^2 + \ldots + \sigma_{sf_1}^2 + \sigma_{sf_2}^2 + \ldots + \sigma_{f_1f_2}^2 + \cdots + \sigma_{res}^2$. Based on this definition, reliability of a performance evaluation across replicated measurements is assessed as the ratio by which the amount of substantial variance outweighs the total error variance. That is, a performance evaluation is deemed reliable if most of the variance is explained by variance between sentences and not by variance within a sentence, such as variance caused by random regularization settings or by residual variance due to unspecified facets of the measurement procedure. Naturally, different assumptions on thresholds on this ratio will lead to different assessments of reliability. A threshold of $80\%$ is used, for example, by Jiang (2018). Values less than $50\%$, between $50\%$ and $75\%$, between $75\%$ and $90\%$, and above $90\%$, are indicative of poor, moderate, good, and excellent reliability, respectively, according to Koo & Li (2016).

VCA and ICCs date back to the works of Fisher (1925). More information can be found in to Chapter 3 and Appendix A2 of Riezler & Hagmann (2022) and Shrout & Fleiss (1979); Searle et al. (1992); McGraw & Wong (1996); Brennan (2001); Webb et al. (2006).

## 5 A Worked-Through Example

We exemplify the methods introduced above on an NLP example from the `paperswithcode.com` open resource, namely the BART+R3F fine-tuning algorithm presented by Aghajanyan et al. (2021) for the task of text summarization, evaluated on the CNN/DailyMail (Hermann et al., 2015) and RedditTIFU (Kim et al., 2019) datasets.

BART+R3F was listed as SOTA for text summarization on these datasets on `paperswithcode.com` at the time of paper publication. It uses an approximate trust region method to constrain updates on embeddings $f$ and classifier $g$ during fine-tuning in order not to forget the original pre-trained representations. This is done by minimizing a task loss $\mathcal{L}(\theta)$ regularized by the Kullback-Leibler distance on normally or uniformly distributed parameters:

$$\mathcal{L}(\theta) + \lambda KL(g \cdot f(x) || g \cdot f(x+z)), \text{ where } z \sim \mathcal{N}(0, \sigma^2 I) \text{ or } z \sim \mathcal{U}(-\sigma, \sigma). \tag{11}$$

The first question we want to answer is that of training reproducibility – is the result difference between baseline and new SOTA reproducible on the data[6] and the code[7] linked on the repository,

---

[5]Among the many advantages of using LMEMs to estimate variance components is that the same model structure can be used for designs that are special cases of the fully crossed design, and the elegant handling of missing data situations. See Baayen et al. (2008); Barr et al. (2013); Bates et al. (2015) for further discussions on the advantages of LMEMs over mixed-model ANOVA estimators.

[6]https://github.com/abisee/cnn-dailymail, https://github.com/ctr4si/MMN

[7]https://github.com/facebookresearch/fairseq/tree/main/examples/rxf

Table 1: Text summarization results (Rouge-1/2/L) for baseline (bl) (BART) and State-of-the-art (SOTA) (BART+R3F) reported in Aghajanyan et al. (2021).

|  | CNN/DailyMail | RedditTIFU |
|---|---|---|
| bl | 44.16/21.28/40.90 | 24.19/8.12/21.31 |
| SOTA | 44.38/21.53/41.17 | 30.31/10.98/24.74 |

Table 2: Significance of result difference baseline-SOTA on CNN/DailyNews under Rouge-1 (R1), Rouge-2 (R2) and Rouge-L (RL) evaluation.

|  | bl | SOTA | $p$-value | effect size |
|---|---|---|---|---|
| R1 | 44.09 | 44.41 | $< 0.0001$ | $-0.101$ |
| R2 | 21.13 | 21.44 | $< 0.0001$ | $-0.080$ |
| RL | 40.81 | 41.16 | $< 0.0001$ | $-0.105$ |

and under the meta-parameter and preprocessing setup reported in the paper. As baseline we take a pre-trained BART-large[8] model (Lewis et al., 2020). The Rouge-1/2/L[9] (Lin & Hovy, 2003) results for the text summarization task reported in Aghajanyan et al. (2021) are shown in Table 1.

Let us first look at the results on the CNN/DailyMail dataset. The paper gives detailed meta-parameter settings for the text summarization experiments, but reports final results as maxima over training runs started from 10 unknown random seeds. Furthermore, the regularization parameter is specified as a choice of $\lambda \in [0.001, 0.01, 0.1]$, and the noise type as a choice from $[\mathcal{U}, \mathcal{N}]$. Using the given settings, we started the BART+R3F code from 5 new random seeds and the BART-large baseline from 18 random seeds on 4 Nvidia Tesla V100 GPUs each with 32 GB RAM and a update frequency of 8. All models were trained for 20-30 epochs using a loss-based stopping criterion. Searching over the given meta-parameter choices, we obtained a training reproducibility result given in Table 2: We find significant improvements of the best SOTA model over the best baseline with respect to all Rouge-X metrics (the difference baseline - SOTA is negative). However, the effect sizes (standardized mean difference between evaluation scores) are small.

Let us next inspect significance conditional on data properties. We quantify properties of summarization inputs by word rarity (Platanios et al., 2019), i.e., the negative logarithm of the empirical probabilities of words in summary inputs, where higher values mean higher rarity. Furthermore, we calculate readability (Kincaid et al., 1975) of summary inputs by calculating the ratio of words/sentences and syllables/word. Readability scores are in principle unbounded, however, an interpretion scheme exists for the range from 0 (difficult) to 100 (easy).

An analysis of significance conditional on data properties can be seen as first step of inferential reproducibility. The interaction plots given in Figure 1 show a significant difference in performance slope for Rouge-2 with respect to ease of readability, where the performance of the best SOTA system increases faster than that of the best baseline for easier inputs (left plot). Also, a significant difference in Rouge-2 with respect to word rarity is seen where the best SOTA model is better than the best baseline for inputs with lower word rarity (right plot).

The next question of inferential reproducibility is whether the results given above are robust against meta-parameter variations, and which meta-parameters are most important in order to achieve the best result. We inspect the original grid of meta-parameter configurations of the SOTA model, given by crossing the given choices of meta-parameters with each other, yielding 3 $\lambda \times$ 2 noise distributions $\times$ 5 random seeds = 30 configurations. As shown in Table 3, the relations between SOTA and baseline are turned around (the difference baseline - SOTA is positive) showing significant wins of baseline over SOTA at medium effect size.

Since the performance variation of the baseline model over 18 random seeds was negligible (standard deviations $< 0.2\%$ for Rouge-X scores), we conduct a reliability analysis of the SOTA model in order to reveal the culprit for this performance loss. The variance component analysis in Table

---

[8]`https://github.com/facebookresearch/fairseq/tree/main/examples/bart`
[9]`https://github.com/google-research/google-research/tree/master/rouge`

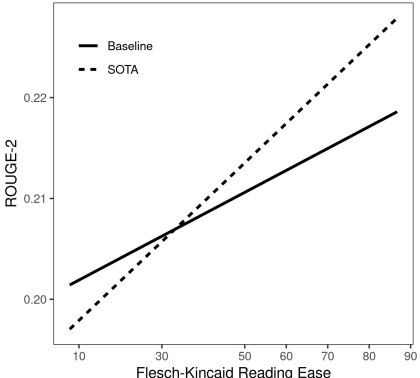 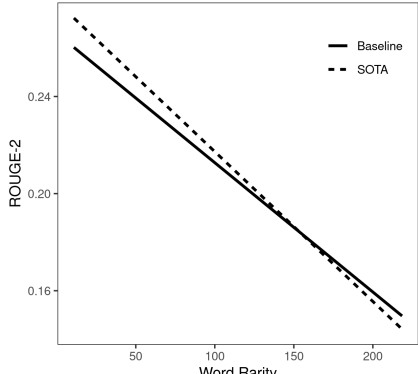

Figure 1: Interaction of Rouge-2 of baseline (solid) and SOTA (dashed) with readability (left) and word rarity (right).

Table 3: Significance of baseline-SOTA on CNN/DailyNews under meta-parameter variation.

|     | baseline | SOTA  | $p$-value  | effect size |
|-----|----------|-------|------------|-------------|
| R1  | 44.15    | 42.21 | < 0.0001   | 0.390       |
| R2  | 21.26    | 19.64 | < 0.0001   | 0.301       |
| RL  | 40.84    | 38.53 | < 0.0001   | 0.531       |

Table 4: Variance component analysis for Rouge-1 (top), Rouge-2 (middle), and Rouge-L (bottom) estimates.

| Variance component $v$ | Variance $\sigma_v^2$ | Percent |
|------------------------|-----------------------|---------|
| summary_id             | 0.00923               | **55.8** |
| lambda                 | 0.00254               | 15.0    |
| random_seed            | 0.00012               | 0.7     |
| noise_distribution     | 0.00005               | 0.3     |
| residual               | 0.00464               | 27.1    |
| summary_id             | 0.00992               | **62.7** |
| lambda                 | 0.00131               | 8.3     |
| random_seed            | 0.00008               | 0.5     |
| noise_distribution     | 0.00003               | 0.2     |
| residual               | 0.00449               | 28.3    |
| summary_id             | 0.00875               | **47.9** |
| lambda                 | 0.00519               | 28.4    |
| random_seed            | 0.00004               | 0.2     |
| noise_distribution     | 0.00001               | 0.1     |
| residual               | 0.00428               | 23.4    |

4 shows that the variance contributions due to variation in random seeds or choice of noise distribution are negligible. However, in all three cases the largest contribution to variance is due to the regularization parameter $\lambda$. The percentage of variance due to objects of interest, here summaries, can readily be interpreted as reliability coefficient $\varphi$, yielding moderate reliability for performance evaluation under Rouge-1 and Rouge-2 ($\varphi$ between 50% and 75%) and poor reliability for evaluation under Rouge-L ($\varphi$ below 50%).

An inspection of the interaction of data properties with the regularization parameter is given in Figure 2. The interaction plots show a significant difference in Rouge-2 performance of the SOTA model between regularization parameters, where performance for $\lambda = 0.1$ is lower and decreases with increasing reading ease (top plot) and increasing word rarity (bottom plot).

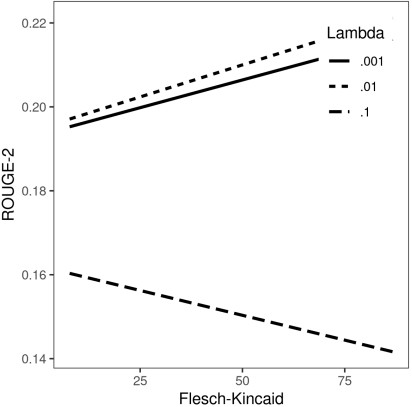 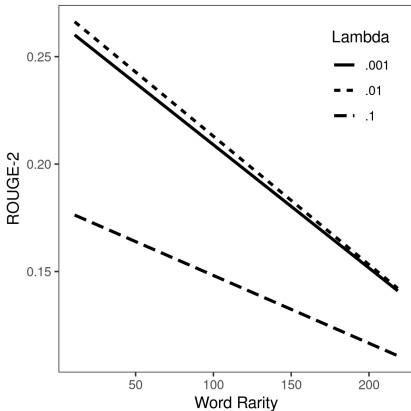

Figure 2: Interaction of Rouge-2 of SOTA for different values of regularization parameter $\lambda$ with readability (left) and word rarity (right).

Let us inspect the results on the RedditTIFU dataset next. These data are interesting since they are much harder to read (mean readability score of $-348.9$), however, a reproducibility analysis on the RedditTIFU dataset was hampered by the fact that the train/dev/test split for RedditTIFU data (long version) was not given on `paperswithcode.com` nor reported in the paper or the code. We used the split[10] provided by Zhong et al. (2020) instead. Under this data split, we found a significant improvement of the best SOTA over the best baseline at a small effect size ($-0.155$) only for Rouge-2. If meta-parameter variation was taken into account, the effect size was even smaller ($-0.0617$). There were no significant interaction effects and neglible variance contributions from meta-parameters.

In sum, this small study allows a nuanced assessment of the strengths and weaknesses of the BART+R3F model: Losing or winning a new SOTA score strongly depends on finding the sweet spot of one meta-parameter (here: $\lambda$), while the paper's goal was explicitly to reduce instability across meta-parameter settings. Performance improvements by fine-tuning are achieved mostly on easy-to-read and frequent-word inputs – these comprise less than one quarter of the CNN/Dailynews data. Furthermore, the optimal choice of main variance-introducing meta-parameter interacts strongly with the investigated data characteristics. Lastly, the model does not seem to be robust against variations in data – under a new random split on RedditTIFU the large gains reported for the split used in the paper can no longer be achieved.

## 6 RELATED WORK

Our work must not be confused with approaches to automatic machine learning (AutoML) Zimmer et al. (2021); Habelitz & Keuper (2020) or neural architecture search (NAS) (Zoph & Le, 2017; Ebrahimi et al., 2017). While AutoML and NAS focus on efficient search of a space of meta-parameter configurations for an optimal result on a particular dataset, our interest is in analyzing the variance contributions of meta-parameters for a given meta-parameter search experiment, and especially in the interactions of meta-parameter variations with characteristics of data, with the goal of gaining insights about possible applications of a model to new data. ANOVA-like techniques have been used to analyze meta-parameter importance in the context of AutoML (Hutter et al., 2014), however, ignoring the crucial aspect of interactions of meta-parameters with data properties. Furthermore, or work is not restricted to meta-parameter variation as source of non-determinism, but can in principle be applied to any source of randomness in machine learning training.

Discussions of reproducibility problems in research date back at least to Ioannidis (2005), and for the area of machine learning at least to Hand (2006). Since then, a multitude of papers has been published on various sources of irreprocucibility in various machine learning areas (see Gundersen et al. (2022) for an overview), however, much less work has been invested in concrete techniques to

---

[10]`https://paperswithcode.com/sota/text-summarization-on-reddit-tifu`

quantify reproducibility. Recent works try to capture reproducibility of evaluations under replicated training runs by a single statistical measure (e.g., coefficient of variation in Belz et al. (2021) or mean and confidence bounds in Lucic et al. (2018); Henderson et al. (2018)), in difference to our goal of decomposing measurement noise into different components and analyzing the interaction of noise factors with data properties. Variance component analysis based on ANOVA techniques has been applied to information retrieval models (Ferro & Silvello, 2016; Voorhees et al., 2017), however, these approaches again ignore an incorporation of data variability into their analysis. We replace ANOVA methods by LMEMs for modeling and estimation (Wood, 2017) and promote the ICC-based idea of quantifying reliability by the proportion of variance attributable to the objects of interest, which to our knowledge has not been applied to machine learning before.

Special-purpose significance tests have been proposed for particular evaluation metrics (Dror et al. (2020), Chapter 3), for meta-parameter variations (Dror et al., 2019), and for multiple test data (Dror et al., 2017). One advantage of the proposed LMEM-based approach is that it unifies these special-purpose techniques into a single framework for hypothesis testing. Furthermore, extensions of bootstrap (Sellam et al., 2022; Bouthillier et al., 2021) or permutation (Clark et al., 2011) tests have been proposed to incorporate meta-parameter variation. The distinctive advantage of our approach is that it enables analyzing significance of result differences conditional on data properties. These can be generic data properties like readability as above, or properties of combined datasets obtained from different sources like data splits, bootstrapped data, or different-domain data sets. The idea of treating test data as random effects and thus increasing the power of statistical significance testing has already proposed by Robertson & Kanoulas (2012). However, the general applicability of LMEMs and GLRTs as a unified framework for significance testing conditional on data properties and simultaneously incorporating arbitrary sources of variation has not yet been fully recognized in the wider community of machine learning research.

## 7 DISCUSSION

Widely recognized work by applied statisticians has proposed to abandon non-confirmatory statistical significance testing, at least its role in screening of thresholds and as guarantor of reproducibility, but instead to report continuous $p$-values, along with other factors such as prior evidence (if available) (Gelman & Loken, 2014; Colquhoun, 2017; McShane et al., 2019). Our proposed use of GLRTs, VCA and ICCs aligns with these recommendations. Our focus is to use them as analysis tools of assess performance under different meta-parameter settings, dependent on characteristics of data, and to detect important sources of variance and their interactions with data properties. This allows us to address questions of genuine interest to researchers and users like "Will the SOTA algorithm's stellar performance on the benchmark testdata lead to top performance on the kinds of datasets that my customers will bring?", or more specifically "How will individual test example characteristics or particular meta-parameter settings, and their interaction with data properties, affect performance?"

Like related work that analyzes model performance under given meta-parameter configurations (Dodge et al., 2019; Strubell et al., 2019), our work is limited by the lack of standardization in the notion of a meta-parameter search. This includes human factors regarding an unclear differentiation between meta-parameters and fixed design choices for particular models. Furthermore, formal criteria on how proper ranges across model families should be defined, are lacking. Thus our work should be seen as a contribution to interpretability of deep learning experiments, not as the provision of a new decisive criterion to rank machine learning models with respect to reliability or a $p$-value under variability of meta-parameters and data. Nonetheless, our methods are readily applicable to performance evaluation data already obtained during meta-parameter optimization. They allow us to transform this usually unused data into new findings about algorithm behavior. We believe that they will be especially useful for large-scale experiments where a manual inspection of variance due to interactions of large numbers of meta-parameters and data properties is prohibitive.

## ACKNOWLEDGEMENTS

This research has been conducted in project SCIDATOS (Scientific Computing for Improved Detection and Therapy of Sepsis), funded by the Klaus Tschira Foundation, Germany (Grant number 00.0277.2015).

ETHICS STATEMENT

The experiments reported in this paper are replications of published results and are not expected to raise any ethical concerns.

REPRODUCIBILITY STATEMENT

The data, code, and meta-parameter settings for the experiments reported in this paper are documented therein and are publicly available.

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
