# OpenReview forum: "Towards Inferential Reproducibility of Machine Learning Research"
_ICLR.cc/2023/Conference — ICLR 2023 poster_

### Official Review · Reviewer_V9qk · 2022-10-15

**Confidence:** 4
**Correctness:** 4
**Technical Novelty And Significance:** 2
**Empirical Novelty And Significance:** 3
**Recommendation:** 8

**Clarity, Quality, Novelty And Reproducibility:**

I think the paper is well structured, organized and written, at the same time its' quality is very good in my opinion.
The paper is novel in terms of the fact it proposes linear mixed effect models that suit much better than standard linear models and AVOVA models to answer the research questions tackled in the research study.
Assessing reproducibility of a paper which is about reproducibility, in particular inferential reproducibility, can seen ilarious, but ... having say that I answer the paper is in principle reproducible by making availabe data, cose and meta-learning parameters used by numerical experiments.
However, I think it would be clearer to refer to the comparison of the performance of two algotihms, rather than using the term SOTA and baseline, I think it would be clearer to talk about SOTA compared to a new algrithm.
The quality and readability of Figure 1 can be improved by making more clear to the interested reader the message it conveys.


**Strength And Weaknesses:**

Strengths:
- the problem addressed is a relevant one, I would say a fundamental problem for good scientific discussion in this times where almost each new pulished paper achieves better performances than well established existing baselines
- the proposed model extends or let say replaces existing models aimed to the same goal
- the proposed approach is elementary and can be easily interpreted and understood, it's assumptions and its' limitations
- the paper is well organized and structured and it read easily and quite well

Weaknesses:
- numerical experiments are limited to only to two datasets, even if this is a weackness of the paper, I think it is a minor one
- I personally do not like the strict statement that reproducibility is the wrong question to ask raher than inferential reproducibility, reproducibility is of great urgency and relevance while inferential reproducibility is a relatively new and fresh concept which deserves attention
- stating that deep learning is non deterministic is a little misleading in my humble opinion, the outcome of learning deep learning models due to many hidded factors turns out to be stochastic but there are no theoretical reason that deep learning models are sthocastic.
- at a give point of page 2, reading the paper, it seems that the authors could mention the no free lunch theorem while they did not, thus I would like them to check whether a link between that important theorem and what they wrote "On the one hand, a successful duplication
of a SOTA result on a benchmark does not guarantee generalization to new data. On the other hand, a model might generalize well to new data even if the exact SOTA result cannot be duplicated because of differences in computational budget."
- maybe providing some more details about experimental design methodology for those who are not aware could improve the effectiveness of the paper, I personally know about about that but I do not think design of experiemnts (DoE) is a methodology that spread in the macine learning community.

**Summary Of The Paper:**

The paper tackles the problem experiments reproducibility with specific reference to inferential reproducibility which proposes to interpret the variation of performance values as due to the following factors; data characteristics, meta-parameter settings, includng also their interactions. The paper starts from the rationale that variance in performance values of different deep learning models and/or machine learning models in general is an intrinsic and interesting effect of non-determinism and a bug to be methodologically solved by researchers. A statistical model, i.e., linear mixed effects model, tailored to analyze performance data from paired experiments (models in this case) is developed to compare performances of state of the art models to that achieved by baseline models. The paper develops the argument of inferential reproucibility and the linear mixed effects model together with numerical experiments on natural language data that show how the proposed approach for inferential reproducibility can be effectively used to state whether the achieved performances by the baseline and the state of the art model are different or not.


**Summary Of The Review:**

The paper is about a fundamental and urgent issue in scintific research, specially under the current setting where it seems that each new method is better and sometimes much better that existing baseline methods.
The paper also introduces and describes the use of linear mixed effect models to compare and futher analyze the perfoamnce achived by state of the art methods and that achieved by baseline methods.
The paper motivates the advantage of the proposed approach in theoretical terms and in a conving manner in my humble opinion.
Furthermore, some numerical experiments are presented to support the statements made by the paper. However, I would have liked to see more numerical experiments to better explain the different aspects of the proposed approach.

---

> ### Author Response · Authors · 2022-11-14
> **Reply to reviewer by authors**
>
> Thank you for your insightful comments. We have uploaded an updated version of the paper that includes clarifications to the following points raised in your review:
>
> - ***Stochasticity of deep learning:*** Stochasticity in weight initialization, dropout, data shuffling and batching is only one factor for non-determinism of deep learning. Other sources of variability in model performance include variations of optimizers, programming code,  data preprocessing, computational infrastructure or computational budget.
>
> - ***"No free lunch" theorem:*** It is tempting to cite Wolpert (1996) in context of dependence of generalization error on data, however, our work does not discuss generalization error in a strict sense. Furthermore, the "no free lunch" theorems are worst-case results whereas we aim at a characterization of reproducibility under sources of variation given by a particular meta-parameter search and given data characteristics.
>
> - Comments on readability: We appreciate your comments on how to make the paper more readable and tried to include them in the updated version of the paper.
>
> Wolpert, David H. (1996). The Lack of a Priori Distinctions between Learning Algorithms. Neural Computation 8, 1341-1390.

---

> > ### Comment · Reviewer_V9qk · 2022-11-14
> > **Confirm that I ...**
> >
> > I confirm that this is not stochasticity, it is simply that you are running different programs on different machines and thus deep learning is not stochastic at all, eventually you do not access all data and informations which could deterministically tell the value of the performance. In other terms it is our ignorance that you call stochasticity but the system itself is still deterministic. The alternative is that you state your system suffers non only epistemic uncertainty but also aleatoric uncertainty. Will appreciate very much you comment on this.

---

### Official Review · Reviewer_yHxA · 2022-10-30

**Confidence:** 3
**Correctness:** 4
**Technical Novelty And Significance:** 2
**Empirical Novelty And Significance:** 2
**Recommendation:** 5

**Clarity, Quality, Novelty And Reproducibility:**

Overall, the paper is well written and clear.
The approaches proposed are not novel, yet application of them to machine learning model comparison is a novel departure from the current state of practice.

**Strength And Weaknesses:**

The authors approach attempts to address criticisms in reproducibility in the machine learning field. By using standard statistical analyses that separate out sources of variation, the authors are able to better characterize in what way the performance of two models may differ. This can increase the understanding in the community of the strengths and weaknesses of different models.

The perspective of the authors is valuable in that it addresses overly simplistic understanding of performance of machine learning models. Namely, one model (SOTA) being categorically better than another (baseline), is an overly simplistic view. Further, the authors' perspective calls into direct question the role of hyper-parameter tuning, random weight initialization, and the availability of computational resources, most of which are not explicitly considered in performance comparisons.

Unfortunately, I do not think the authors have presented enough study of their point of view for a viable paper. While the authors propose to improve upon the comparison of models, their enumeration of the many dimensions of flexibility in their approach makes it clear that such an approach could easily exacerbate issues of uncertainty rather than resolve them. While being flexible enough to include many different sources of variation is useful, a high degree of standardization of the comparison is needed. Further, the authors do not address the power of their proposed test.

In the end, one could argue that model comparison has more to do with the ability of a model to perform well on novel problems, beyond the test set used for the comparison. One chooses to use a SOTA model not because it performed well on some historical test set, but because one hopes that it will perform well on one's problem at hand. The method proposed by the authors does address this in some capacity by quantifying the variation due to observations (sentences) separately from that due to hyper-parameters and random initialization.

More standardization and demonstration of the proposed approach are needed.
While the authors mention that code is available, they do not really specify what that code does. Libraries that make it easy for authors to use the proposed approach, would be welcome.

**Summary Of The Paper:**

The authors propose the use of nested linear mixed effects models to compare performance of machine learning models. This approach is proposed to incorporate the effects of random initialization, hyper-parameter values, and other non-deterministic elements of model evaluation in the comparison. The authors then detail a reliability coefficient that compares the amount of variance in model performance due to the above aspects to the total variance. This enables the quantification of the notion of reliability in machine learning model evaluation.

**Summary Of The Review:**

A great start, but this paper needs more.

---

> ### Author Response · Authors · 2022-11-14
> **Reply to reviewer by authors**
>
> Thank you for your insightful comments. We have uploaded an updated version of the paper that includes clarifications to the following points raised in your review:
>
> - ***Standardization:*** As mentioned in the general comment, we do not claim to provide a decisive test to rank model performance under variability of meta-parameters and data. The main purpose of a significance tests in our work is as an analysis tool detect cases where significance is lost, rather than as a decisive test to verify significance under meta-parameter variation. The same holds for tools for variance component analysis: We do not use them to threshold reliability, but rather as tools to analyze the sources of variance and their interactions with data properties.
>
> - ***Performance on novel problem:*** We fully subscribe to your interpretation our our work, stating that "in the end, one could argue that model comparison has more to do with the ability of a model to perform well on novel problems, beyond the test set used for the comparison. One chooses to use a SOTA model not because it performed well on some historical test set, but because one hopes that it will perform well on one's problem at hand." This is the contribution of analyzing variation due to data characteristics. An analysis of variance conditional on data properties is much less affected by standardization problems mentioned above.
>
> - ***Code availability:*** Code (R and Python) for the toolkit and several sample applications are part of the original submission as compressed repository in the supplementary material. Please have look! A link to a website hosting code and data will be made available upon publication of the paper.

---

### Official Review · Reviewer_994g · 2022-10-31

**Confidence:** 4
**Correctness:** 3
**Technical Novelty And Significance:** 3
**Empirical Novelty And Significance:** 2
**Recommendation:** 5

**Clarity, Quality, Novelty And Reproducibility:**

The paper is clear and appears to be reproducible. What is less clear is novelty. See above comments for details.

**Strength And Weaknesses:**

\+ The paper discusses an extremely important issue around generalization and statistical variability. This issue has a long history and a central role in science (see below), but that understanding appears to be missing form many current ML researchers. This paper is a useful remedy for that problem.

– The authors make a high-level point about characterizing and comparing variability which is a longstanding issue in science generally and statistics specifically. Then they recommend the use of several known methods in statistics. Thus, it is unclear what is novel about the paper. The authors appear to contend that researchers in deep learning are not making sufficient use of these ideas and methods (and I would tend to agree). However, there is little evidence of that point in the current version of the paper.

– The paper does not cite some relevant scholarship in this area, including Andrew Gelman's work on "the garden of forking paths" (Gelman & Loken 2014) and David Hand's work on "the illusion of progress" (Hand 2006) in the context of classification. Others (e.g., Clary et al. 2019; Cobbe et al. 2019; Agarwal 2021) have raised issues of variability in reporting of results of deep learning methods.

– The authors present their view as "heretical" when a long tradition of work exists on how to make inferences in the face of statistical variability. That some researchers in machine learning appear to have missed these basic methods of science hardly make the authors' viewpoint heretical.

– The authors claim to present "...a new type of reproducibility called inferential reproducibility..." They note that "It asks the following question: Can qualitatively similar conclusions be drawn from an independent replication of a study?" Here, the authors are identifying the difference between what is sometimes called "reproducibility" (different team; same experimental setup) and "replicability" (different team; different experimental setup) (ACM 2020). They, themselves, note some prior scholarship on this issue, so it is unclear why they refer to it as "a new type of reproducibility."

– The authors describe a general problem (a focus on replicating precise values of performance rather than assessing and understanding the effects of variability in performance), and then jump immediately to specific statistical methods. An intermediate topic appears to be entirely missing: What sorts of generalization are being evaluated in a given study, what sorts of generalization are to be reasonably expected in practical uses, and how do those two match up? For example, Hand (2006) explicitly identifies explicit differences of the joint distribution between training and test as likely in realistic cases, and he conjectures that newer methods for classification are likely to be more sensitive to such differences. Similarly, work in concept drift, transfer learning, and related fields have assessed the robustness of specific types of machine learning models to specific types of differences in the distributions and when those sorts of differences are to be expected in practice. The paper would be improved by discussing this sort of middle-ground issue that connects the high-level points of the introduction and the specific statistical methods proposed for use.

References

Association for Computing Machinery (2020). Artifact Review and Badging. Version 1.1.

Agarwal, R., Schwarzer, M., Castro, P. S., Courville, A. C., & Bellemare, M. (2021). Deep reinforcement learning at the edge of the statistical precipice. Advances in neural information processing systems, 34, 29304-29320.

Clary, K., Tosch, E., Foley, J., & Jensen, D. (2019). Let's Play Again: Variability of Deep Reinforcement Learning Agents in Atari Environments. arXiv preprint arXiv:1904.06312.

Cobbe, K., Klimov, O., Hesse, C., Kim, T., & Schulman, J. (2019, May). Quantifying generalization in reinforcement learning. In International Conference on Machine Learning(pp. 1282-1289). PMLR.

Gelman, A., & Loken, E. (2014). The statistical crisis in science data-dependent analysis—a “garden of forking paths”—explains why many statistically significant comparisons don’t hold up. American scientist, 102(6), 460.

Hand, D. J. (2006). Classifier technology and the illusion of progress. Statistical science, 21(1), 1-14.

**Summary Of The Paper:**

The authors propose an approach to reproducibility that focuses on assessing and reporting variability in performance rather than focusing on replicating precise performance values.  The authors propose using a linear mixed-effects model to analyze the factors that influence performance.

**Summary Of The Review:**

An important topic, and lots of interesting exposition about the high-level problem and specific statistical methods. However, the novelty is less clear.

---

> ### Author Response · Authors · 2022-11-14
> **Reply to reviewer by authors**
>
> Thank you for your insightful comments. They forced us to clarify several important aspects in the paper. We have uploaded an updated version of the paper that includes clarifications to the following points raised in your review:
>
> - ***Novelty:*** As stated in the general comment, the main contribution of the paper is a tool to analyze inferential reproducibility of machine learning experiments. The tool is based on well-known statistical techniques for variance component analysis and significance testing, however, to our knowledge they are for the first time applied to untangle multiple sources of variability in performance of machine learning models. We focus on performance variability introduced by meta-parameter settings and data characteristics. The goal is to quantify the effect of non-determinism in machine learning on performance variability, and to analyze and understand the contribution of meta-parameter settings and data characteristics.
>
> - ***Related work:*** We appreciate your pointers to related work. We included a discussion of the role of p-values in the reproducibility crisis, citing seminal works like Ioannidis 2005, Hand 2006, or Gelman 2014. Since it is impossible to cite every paper that reports a new aspect of reproducibility on a new machine learning task,  we tried to cite the first (or very early) papers that address particular aspects in the respective discussion, without aiming at completeness. For example, for early work that shows how non-determinism in deep learning can reverse the relations between baseline and SOTA, we cited Melis et al. 2018; for work that advocate to compare distributions of performance scores instead of point estimates; we cited Reimers et al. 2017; for work that advocates to report confidence intervals, we cited Henderson et al. 2018.
>
> - ***Contribution to theoretical discussion:*** As stated in the general comment, the paper does not claim contributions to the theoretical discussion of different notions of reproducibility/replicability/repeatability. We made it clear or work builds on the notion if "inferential reproducibility" that has been introduced by Goodman et al. (2016). We cited Plesser 2018 who discusses the confused terminology and shows how Goodman's "inferential reproducibility" falls outside the scope of the standard discussion in that any kind of variability is allowed, for example, different data, different meta-parameters, different programming code, different computing infrastructure, different researchers, etc.

---

### Author Response · Authors · 2022-11-14
**General comments to resolve misunderstandings in the reviews**

We have uploaded an updated version of the paper that includes clarifications on the following points:

- The ***novelty and main contribution of the paper is a tool to analyze inferential reproducibility of machine learning experiments***. The tool is based on well-known statistical techniques for variance component analysis and significance testing, however, to our knowledge they have not been applied to machine learning experiments before. The goal is to quantify the effect of non-determinism in machine learning on performance variability, especially, to  analyze and understand the variance contribution of meta-parameter settings and data characteristics.

- The paper does ***NOT claim contributions to the theoretical discussion*** of different notions of reproducibility/replicability/repeatability or of different aspects of generalization. Instead, we start from the general notion of "inferential reproducibility" that has been introduced by Goodman et al. (2016). It extends related notions by allowing any kind of variation in an experiment. For machine learning experiments, sources of variance can be different data, different meta-parameters, or others facets like different programming code, different computing infrastructure, different researchers, etc.

- We also do ***NOT claim to provide a decisive test to rank machine learning models with respect to performance under variability of meta-parameters and data***. Reliability coefficients and analysis of variance are meant as analysis tools that allow to pinpoint the sources contributing to performance variation. Also, the role of p-values and significance testing in our work is meant as an analysis tool, not as a decisive instrument for reproducibility.

Steven N. Goodman, Daniele Fanelli, and John P. A. Ioannidis. What does research reproducibility mean? Sci Transl Med, 8(341):1–6, 2016.

---

### Decision · Program_Chairs · 2023-01-20

**Decision:**

Accept: poster

**Justification For Why Not Higher Score:**

Average reviewers' score not high enough, even though the work is intriguing.

**Justification For Why Not Lower Score:**

A thought provoking work that targets an important practical issue and deserves to be presented at ICLR.

**Metareview: Summary, Strengths And Weaknesses:**

This submission has been assessed by three knowledgeable reviewers. Two of them rated it marginally below the acceptance threshold, one gave it a strong accept. It is indeed a thought-provoking paper and as such, in my opinion, it should be presented at ICLR. It targets an important topic and even though, as the reviewers correctly point out, there is no novelty in the approaches used (their key complaint), but their application to comparison of machine learning models is.

**Note From Pc:**

if the above contains the word "oral" or "spotlight" please see: "oral" presentation means -> notable-top-5% and "spotlight" means -> notable-top-25%. As stated in our emails, we are disassociating presentation type from AC recommendations